# MULTIPLICATIVE INTERACTIONS
# AND WHERE TO FIND THEM

**Siddhant M. Jayakumar, Wojciech M. Czarnecki, Jacob Menick, Jonathan Schwarz,
Jack Rae, Simon Osidnero, Yee Whye Teh, Tim Harley, Razvan Pascanu**
DeepMind
```
{sidmj, lejlot, jmenick, schwarzjn, jwrae, osindero,
ywteh, tharley, razp}@google.com
```

## ABSTRACT

We explore the role of multiplicative interaction as a unifying framework to describe a range of classical and modern neural network architectural motifs, such as gating, attention layers, hypernetworks, and dynamic convolutions amongst others. Multiplicative interaction layers as primitive operations have a long-established presence in the literature, though this often not emphasized and thus under-appreciated. We begin by showing that such layers strictly enrich the representable function classes of neural networks. We conjecture that multiplicative interactions offer a particularly powerful inductive bias when *fusing multiple streams of information* or when *conditional computation* is required. We therefore argue that they should be considered in many situation where multiple compute or information paths need to be combined, in place of the simple and oft-used concatenation operation. Finally, we back up our claims and demonstrate the potential of multiplicative interactions by applying them in large-scale complex RL and sequence modelling tasks, where their use allows us to deliver *state-of-the-art* results, and thereby provides new evidence in support of multiplicative interactions playing a more prominent role when designing new neural network architectures.

## 1  INTRODUCTION

Much attention has recently turned toward the design of custom neural network architectures and components in order to increase efficiency, maximise performance, or otherwise introduce desirable inductive biases. While there have been a plethora of newer, intricate architectures proposed, in this work we train our sights instead on an older staple of the deep learning toolkit: multiplicative interactions.

Although the term itself has fallen somewhat out of favour, multiplicative interactions have reappeared in a range of modern architectural designs. We start this work by considering multiplicative interactions as an object of study in their own right. We describe various formulations and how they relate to each other as well as connect more recent architectural developments (e.g. hypernetworks Ha et al. (2017), dynamic convolutions Wu et al. (2019)) to the rich and longer-standing literature on multiplicative interactions.

We hypothesise that multiplicative interactions are suitable for representing certain meaningful classes of functions needed to build algorithmic operations such as conditional statements or similarity metrics, and more generally as an *effective way of integrating contextual information in a network* in a way that generalizes effectively. We show this empirically in controlled synthetic scenarios, and also demonstrate significant performance improvement on a variety of challenging, large-scale reinforcement learning (RL) and sequence modelling tasks when a conceptually simple multiplicative interaction module is incorporated.

Such improvements are consistent with our hypothesis that the use of appropriately applied multiplicative interactions can provide a more suitable inductive bias over function classes leading to more data-efficient learning, better generalization, and stronger performance. We argue that these operations should feature more widely in neural networks in and of themselves, especially in the

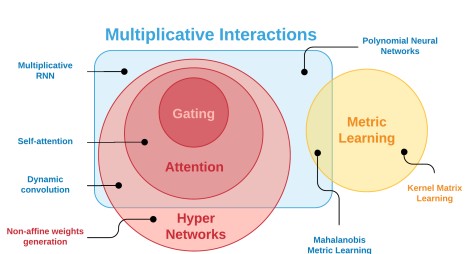
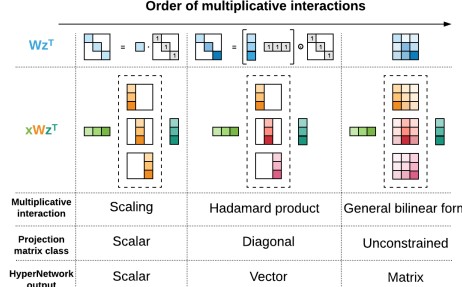

Figure 1: (Left) Venn diagrams of multiplicative interactions with respect to other model classes commonly used in ML. (Right) Comparison of various orders of multiplicative interactions and their relation to other perspectives.

increasingly important setting of *integrating multiple streams of information* (including endogenously created streams e.g. in branching architectures).

Our contributions are thus: (i) to re-explore multiplicative interactions and their design principles; (ii) to aid the community's understanding of other models (hypernetworks, gating, multiplicative RNNs) through them; (iii) to show their efficacy at representing certain solutions; and (iv) to empirically apply them to large scale sequence modeling and reinforcement learning problems, where we demonstrate *state-of-the-art* results.

## 2 MULTIPLICATIVE INTERACTIONS

We start by introducing notation and formalising the concept of *multiplicative interactions*. The underlying question we are trying to answer is how to combine two different streams of information. Specifically, given $\mathbf{x} \in \mathbb{R}^n$ and $\mathbf{z} \in \mathbb{R}^m$, our goal is to model an unknown function $f_{\text{target}}(\mathbf{x}, \mathbf{z}) \in \mathbb{R}^k$ that entails some interaction between the two variables. In practice $\mathbf{x}$ and $\mathbf{z}$ might be arbitary hidden activations, different input modalities (e.g. vision and language), or conditioning information and inputs.

The standard approach is to approximate $f_{\text{target}}$ by a neural network $f$. If $f$ is restricted to employ a single layer of weights, we typically use $f(\mathbf{x}, \mathbf{z}) = \mathbf{W}[\mathbf{x}; \mathbf{z}] + \mathbf{b}$, where $[\mathbf{x}; \mathbf{z}]$ represents the concatenation of $\mathbf{x}$ and $\mathbf{z}$, and $\mathbf{W} \in \mathbb{R}^{(m+n) \times k}$ and $\mathbf{b} \in \mathbb{R}^k$ are learned parameters. The interaction between $\mathbf{x}$ and $\mathbf{z}$ is only additive given this formulation. However through stacking multiple similar layers (with element-wise nonlinearities inbetween), $f$ can approximate any function $f_{\text{target}}$ given sufficient data (and capacity).

In contrast, a single layer with *multiplicative interactions* would impose the functional form

$$f(\mathbf{x}, \mathbf{z}) = \mathbf{z}^T \mathbb{W} \mathbf{x} + \mathbf{z}^T \mathbf{U} + \mathbf{V} \mathbf{x} + \mathbf{b} \tag{1}$$

where $\mathbb{W}$ is a 3D weight tensor, $\mathbf{U}, \mathbf{V}$ are regular weight matrices and $\mathbf{b}$ is a vector[1]. We posit that this specific form, while more costly, is more flexible, providing the right inductive bias to learn certain families of functions that are of interest in practice. Additionally, many existing techniques can be shown to rely on variations of the above bilinear form as detailed below.

**Hypernetworks as Multiplicative Interactions.** A *Hypernetwork* Ha et al. (2017) is a neural network $g$ that is used to generate the weights of another neural network given some context or input vector $\mathbf{z}$. Particularly $f(\mathbf{x}; \theta)$ becomes $f(\mathbf{x}; g(\mathbf{z}; \phi))$. In the case where $f$ and $g$ are affine (as in the original work), such a network is exactly equivalent to the multiplicative form described above.

Specifically, we can decompose equation (1) and set $\mathbf{W}' = \mathbf{z}^T \mathbb{W} + \mathbf{V}$ and $\mathbf{b}' = \mathbf{z}^T \mathbf{U} + \mathbf{b}$. We can now see $\mathbf{W}'$ as the generated 2D weight matrix and $\mathbf{b}'$ as the generated bias from some hypernetwork.

---

[1]We have the 3D tensor $\mathbb{W} \in \mathbb{R}^{m \times n \times k}$ and $(\mathbf{z}^T \mathbb{W} \mathbf{x})_k = \sum_{ij} \mathbf{z}_i \mathbb{W}_{ijk} \mathbf{x}_j$.

This allows us to have an input-conditional weight matrix and bias vector that are then used to generate output $\mathbf{y} = \mathbf{W}'\mathbf{x} + \mathbf{b}'$. We can also consider the more general case of any affine transformation being generated by some arbitrary neural network, which can also be viewed as a multiplicative interaction where we first embed the context $\mathbf{z}$ and then use it in the equation above. This provides a basis for thinking about hypernetworks themselves as variations on the theme of multiplicative interactions, potentially accounting for a considerable amount of their efficacy.

**Diagonal Forms and Gating Mechanisms.** Let us consider a diagonal approximation to the projected $\mathbf{W}'$. This is given by a particular parametrization of $\mathbf{W}' = \mathbf{z}^T\mathbb{W} + \mathbf{V}$ above (see Figure 1 right). Multiplying with $\mathbf{W}' = \mathrm{diag}(a_1, ..., a_n)$ can be implemented efficiently as $f = \mathbf{a} \odot \mathbf{x}$ where $\odot$ represents elementwise multiplication or the Hadamard product (similarly for the bias). This form now resembles commonly used gating methods, albeit they are often used with additional non-linearities (e.g. sigmoid units Dauphin et al. (2017); Van den Oord et al. (2016)). It can be viewed as a hypernetwork as well, where $\mathbf{z}^T\mathbf{W}$ represents the function generating parameters.

**Attention and Multiplicative Interactions.** While not the focus of this work, we note that attention systems in sequence modelling (Vaswani et al., 2017; Bahdanau et al., 2014) similarly use multiplicative interactions to effectively scale different parts of the input. This is typically done using the diagonal form above with $\mathbf{m} = f(\mathbf{x}, \mathbf{z}), \mathbf{y} = \mathbf{m} \odot \mathbf{x}$ where $\mathbf{m}$ is often a bounded mask. Attention systems are typically used with different aims to those we describe here: they can suppress or amplify certain inputs and allow long-range dependencies by combining inputs across time-steps (when masking above is followed by a pooling layer, for example). We use these insights to posit that while more expensive, considering a higher order interaction (generating a vector mask) might prove more beneficial to such systems but we do not specifically consider attention in this paper and leave it to future work.

**Scales and Biases.** Further, we can make another low-rank approximation to the diagonal form and generate instead a *scalar matrix* – i.e. the hypernetwork outputs a single *scalar* scale (and/or bias) parameter per channel or feature vector we are considering, instead of a vector. We can again write this as $f = \mathbf{z}^T\mathbb{W} \odot \mathbf{x}$ where $\mathbf{z}^T\mathbb{W} = \alpha\mathbb{I}$. This is common in methods such as FiLM (Perez et al., 2018; Dumoulin et al., 2018; 2017).

**Multiplicative Interaction and Metric Learning.** Another highly related field of active research is that of metric learning, where one tries to find the most suitable metric to measure similarity between objects in some parametrised space of metrics. One of the most commonly used classes is that of Mahalanobis distances $d_{\mathbf{C}}(\mathbf{x}, \mathbf{z}) = \|\mathbf{x} - \mathbf{z}\|_{\mathbf{C}} = (\mathbf{x} - \mathbf{z})^T\mathbf{C}^{-1}(\mathbf{x} - \mathbf{z})$, which again maps onto multiplicative interaction units as $d_{\mathbf{C}}(\mathbf{x}, \mathbf{z}) = (\mathbf{x} - \mathbf{z})^T\mathbf{C}^{-1}(\mathbf{x} - \mathbf{z}) = \mathbf{x}^T\mathbf{C}^{-1}\mathbf{x} - 2\mathbf{x}^T\mathbf{C}^{-1}\mathbf{z} + \mathbf{z}^T\mathbf{C}^{-1}\mathbf{z}$. In metric learning, however, one usually explicitly defines losses over tuples (or higher order n-tuples) with direct supervision, while here we consider building blocks that can learn a metric internally, without direct supervision.

**The Taxonomy of Multiplicative Interactions.** Finally, we summarise these relationships in figure 1. We can think of multiplicative interactions equivalently in terms of either: (a) the approximation to the 3D tensor made; (b) the output of the "projected" context by the hypernetwork; or (c) the operation used to combine the generated weights/context and the input. For example, the general bilinear form is equivalent to a vanilla hypernetwork that generates a weight matrix for a matrix multiplication. Similarly, a diagonal 3D tensor is equivalent to a hypernetwork that generates a vector and is combined with a hadamard product.

## 3 EXPRESSIVITY OF THE MODEL

Vanilla MLPs are universal approximators – that is, for every continuous function $[0, 1]^d \to \mathbb{R}$ (considered our target) and every approximation error $\epsilon > 0$ there exist hidden units $H$ and corresponding parameter values $\theta$ such that the distance in function space between the MLP output and the target function is smaller than $\epsilon$. Consequently adding new modules/building blocks does not affect the approximation power of neural nets, however such modifications *can* change the *hypotheses space* – the set of functions that can be represented exactly (with 0 error), and the compactness of a good estimator (how many parameters are needed), as well as learnability.

We first show that multiplicative interactions strictly enlarge the hypotheses space of vanilla MLPs – that is, we add new functions which multi-layer multiplicative models can now represent perfectly,

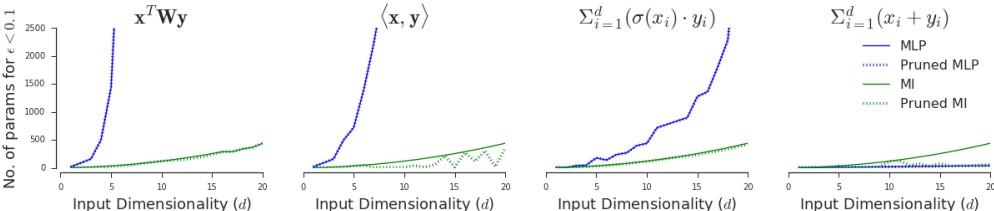

Figure 2: Number of parameters needed for a regular, single layer MLP (blue line) to represent the function up to 0.1 MSE over the domain of a standard $d$-dimensional Gaussian compared to the same quantity for a multiplicative model (green line). $\sigma$ denotes sigmoid. Dotted lines represent pruned models where all weights below absolute value of 0.001 were dropped. Note that for MLP all parameters are actually used, while for MI module some of these functions (summation and dot product) can be compactly represented with pruning.

while also preserving our ability to represent those in the existing set modeled perfectly by vanilla MLPs (full proof in appendix A).

**Theorem 1.** *Let $\mathcal{H}_{mlp}$ denote the hypotheses space of standard MLPs with ReLU activation function, and let $\mathcal{H}_{mu}$ denote the hypotheses space of analogous networks, but with each linear layer replaced with a multiplicative layer, then we have $\mathcal{H}_{mlp} \subsetneq \mathcal{H}_{mu}$.*

While valuable, such a result can also be trivially obtained by adding somewhat exotic activation functions (e.g. Weierstrass function $\sigma(x) = \sum_{n=0}^{\infty} 0.5^n \cos(7^n \pi x)$ which is a continuous function but nowhere differentiable (Weierstrass, 1895); see appendix for the proof) to the pool of typically used ones. While increasing the hypothesis space on its own is not of great significance, the crucial point here is that the set $\mathcal{H}_{mu} \setminus \mathcal{H}_{mlp}$ helps extend our coverage to the set of basic functions that one would expect to need in composing solutions that mimic systems of interest – such as logical, physical, or biological ones.

Figure 2 shows the learnability (up to a certain error) of some simple two input functions against the number of parameters needed. We consider summation, gating, and dot products – which are basic buildings blocks of operations such as conditional statements or similarity metrics, and fundamental for implementing rich behaviours when combining different sources of information. For the gating and dot-product function classes, the complexity of MLPs required to learn them seems to grow exponentially, while the growth for multiplicative models is quadratic. On the other hand summation is trivially easier for an MLP. Thus we do not argue that multiplicative interactions are a silver bullet – but that such interactions add an important class of functions to the hypothesis set that are often the right inductive bias, or algorithmic building block, for many kinds of problems. In subsequent sections we show empirically that using them as *context-integration layers* leads to good performance gains across a range of tasks.

## 4 RELATED WORK

There is a vast body of literature surrounding multiplicative interactions, and these ideas have a long history, for example being discussed in the foundational era of connectionism (Rumelhart et al., 1986). Below we highlight some of the key works developed in the community over the last few decades and aim to show how these relate to each other.

Some of the earliest models leveraging multiplicative interactions were higher-order Boltzmann machines or autoencoders (Sejnowski, 1986; Memisevic & Hinton, 2007; Taylor & Hinton, 2009). Currently, the most common usage of multiplicative interactions seen in models that enjoy widespread adoption is via a factorised or diagonal representation of the necessary 3D weight tensor. The LSTM cell (Hochreiter & Schmidhuber, 1997) (and its descendents such as the GRU (Cho et al., 2014)) employ multiplicative interactions of this form in the gating units that are crucial for the long-term stability of memories. Enhanced multiplicative versions of LSTMs have also been formulated (Sutskever et al., 2011; Wu et al., 2016; Krause et al., 2016): these approaches essentially combine the previous hidden state and current input via an element-wise or hadamard product between projected representations.

Similarly, bilinear layers (often low-rank factorizations) have appeared extensively in the computer vision literature (Gao et al., 2016; Kim et al., 2016) and beyond (Dumoulin et al., 2018). Squeeze-and-excitation networks, for example, can be seen as an instantiation of this idea (Hu et al., 2018). Specifically in visual-question answering systems, models like FiLM (Perez et al., 2018) or class-conditional batch norm (Brock et al., 2019; Perez et al., 2017) use such diagonal forms to generate per-channel scales and biases as a function of some context. This has been shown to be effective at capturing relationships between the two different modalities (text and vision), as well as providing a powerful mechanism to allow a single network to conditionally specialize on multiple different tasks. Further, multimodal domains such as VQA have also seen such bilinear models used in combination with attention systems (Yu et al., 2017; Lu et al., 2016; Xu & Saenko, 2016; Schwartz et al., 2017).

Further, there are many additional works using gating mechanisms which can be thought of as such diagonal approximations used in conjunction with additional point-wise non-linearities or softmaxes. Recent examples of such include pixelCNNs (Van den Oord et al., 2016) and Highway Networks (Srivastava et al., 2015; Zilly et al., 2017), among others (Dauphin et al., 2017), and earlier examples can be seen in works such as Mixtures of Experts (Jacobs et al., 1991) and successors.

Multiplicative interactions in the non-factorised sense can also be thought of as a restricted class of Hypernetworks (Ha et al., 2017): models that generate the weights of one network from another. While the original presentation (Ha et al., 2017) considered their use for model compression in feed-forward nets (i.e. using layer IDs to generate weights), they also investigate HyperLSTMs, in which per timestep multiplicative biases are generated. A similar approach has also been applied to generating parameters in convolutional nets via "dynamic convolutions" where the size of the generated parameters is controlled by tying filters (Wu et al., 2019). Further, these ideas have been extended to Bayesian forms (Krueger et al., 2017) and also used for example, in architecture search (Brock et al., 2017).

Lastly, multiplicative interactions used to scale contributions from different spatial or temporal components play a key role in attention mechanisms (Bahdanau et al., 2014; Vaswani et al., 2017). They have also been used in some RL works to better condition information, e.g. in Feudal Networks (Vezhnevets et al., 2017) as a way for manager and worker units to interact, and better action-conditioning (Oh et al., 2015).

## 5 EXPERIMENTAL SETUP

We aim to demonstrate that the incorporation of multiplicative interactions can boost performance across a wide range of problems and domains, and we conjecture that this is because they effectively allow for better routing and integration of different kinds of information. Specifically we will show that multiplicative interactions allow better integration of *(a) latent variables* in decoder models, *(b) task or contextual information in multitask learning*, *(c) recurrent state* in sequence models. We use neural process regression, multitask RL and language modelling as exemplar domains. Further details of architectures and hyper-parameters can be found in the appendix.

**A Note on Implementation and Terminology.** We use $\mathcal{M}(\mathbf{x}, \mathbf{z})$ below to mean the function $f(\mathbf{x}, \mathbf{z}) = \mathbf{z}^T \mathbb{W} \mathbf{x} + \mathbf{z}^T \mathbf{U} + \mathbf{B}\mathbf{x} + \mathbf{b}$ (or referred to as MI in the legends). In all cases we implement this using a series of standard linear layers with a reshape operation in between to form the intermediate matrix (equivalently this can be done with einsum or tensor product notation; we provide a simple implementation in the appendix). The quantity $f_1(\mathbf{z}) = \mathbf{z}^T \mathbb{W} + \mathbf{B}$, (where as above $\mathbb{W}$ is 3D and $\mathbf{B}$ a 2D bias) represents the 2D output of projecting the contextual information. We refer to this interchangeably as the 2D-contextual projection or "generated weights" using the hypernet terminology. Similarly the "generated bias" is the 1D projection of the context $\mathbf{z}$, that is, $f_2(z) = \mathbf{z}^T \mathbf{U} + \mathbf{b}$ (and thus $f(\mathbf{x}; \mathbf{z}) = f_1(\mathbf{z})\mathbf{x} + f_2(\mathbf{z})$).

While we aim to draw connections to other models and approximations in the literature above, our aim is not to advertise one specific instantiation or approximation over the other . As such, we use the same form above in all experiments (unless specified otherwise) and control parameter count by controlling the size of $\mathbf{z}$. Undoubtedly, practitioners will find that task-specific tuning of the above form might yield comparable or better results with fewer parameters, given the right approximations.

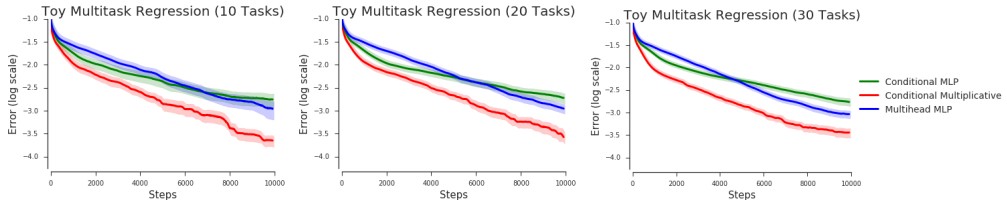

Figure 3: Averaged learning curves for different models while varying the number of tasks in the toy multitask regression domain. Shaded regions represent standard error of mean estimation.

## 6 LEARNING CONTEXT DEPENDENT LAYERS FOR MULTITASK LEARNING

We start by considering the general paradigm of multitask-learning, wherein the goal is to train one model to solve $K$ different tasks. We show that we can boost performance here by learning context or task-conditional layers with multiplicative interactions. There is generally a trade-off between the transfer from similar tasks and the negative interference between those with very different solutions. Our claim is thus that context-conditional layers provide a best-of-both-worlds approach, with an inductive bias that allows transfer (as opposed to a multiheaded architecture) while also limiting interference.

We first demonstrate this with a toy example, where we attempt to regress two different classes of functions, affine and sines, with one model. Specifically, $y = a_i x + b_i$ and $y = a_i \sin(10x) + b_i$ where $a_i$ and $b_i$ are sampled per task from a uniform distribution. In Figure 3 we show results averaged over multiple runs, as we vary the number of tasks. We train both a standard MLP with multiple heads and one with that is given task ID as additional input and can see that the neither is able to use task information to do any better. On the other hand, the results show that a task-conditioned $\mathcal{M}(x, \mathbf{t})$ layer allows the model to learn both tasks better with less interference and also increased data efficiency. We see that the gains with using an $\mathcal{M}$ layer are more pronounced as we increase the number of tasks. More details are provided in appendix D.

### 6.1 MULTITASK RL ON DMLAB-30

Next, we consider a larger scale problem: multitask RL on the DeepMind Lab-30 domain (Beattie et al., 2016). This is a suite of 30 tasks in a partially-observable, first-person-perspective 3D environment, encompassing a range of laser-tag, navigation, and memory levels. We use a typical actor-critic RL setup within the Impala framework of Espeholt et al. (2018), with multiple actors and a single learner with off-policy correction (further details provided in appendix section E). We use exactly the architecture as in the original works (Espeholt et al., 2018; Hessel et al., 2019): a stack of convolutional layers followed by an LSTM, with output $\mathbf{h}_t$ at each timestep. Normally these are then projected to policies $\pi$ and value functions $V$ that are shared across all tasks. At test time agents are typically not allowed to access ground truth task ID (e.g. this means value functions at training time could use this information).

**Multi-head Policy and Value Layers** We first show that a multi-headed agent architecture with one policy and value head per level does in fact boost performance. While this does use privileged information (task ID), this does show that there is some degree of interference between levels from sharing policy and value layers.

**Multiplicative Policy and Value Layers** We can instead consider using multiplicative layers here to integrate task information (one-hot task ID $\mathbf{I}_i$) to modulate compute-paths. We learn a task embedding as below, and use it in a multiplicative layer that projects to policy and value functions. That is, we now have $\mathbf{c} = \mathrm{relu}(\mathrm{MLP}(\mathbf{I}_i))$ as a context, and $\pi_t, V_t = \mathcal{M}(\mathbf{h}_t, \mathbf{c})$.

We show results in the same figure and find that such a layer provides a further boost in performance. This can be viewed as generating policy layers from some learnt embedding of the task ID. Our hypothesis is that while multiheaded architectures reduce interference, they also remove the ability of policy-transfer between the different layers. Further, each head only gets $1/K$ of the number of gradient updates (when training on $K$ tasks).

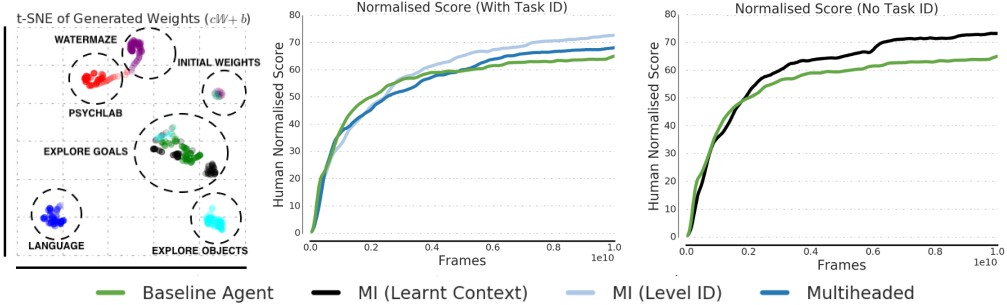

Figure 4: (a) A t-SNE plot of the generated weights from an $\mathcal{M}$ layer. (b) Human normalised performance (capped at 100) when using task ID as context to an $\mathcal{M}$ layer. (c) Using a learnt context instead.

**Multiplicative Policies with *Learnt* Contexts** We find (somewhat surprisingly) that we can get similar or greater performance gains *without* using any task information, replacing task ID $\mathbf{I}_i$ instead with a learnt non-linear projection of the LSTM output. We now have, $\mathbf{c} = \text{relu}(\text{MLP}(\mathbf{h}_t))$ and $\pi_t, V_t = \mathcal{M}(\mathbf{h}_t, \mathbf{c})$.

We show a t-SNE plot of the 2D projection of the state of the LSTM in Figure 4. This is coloured by level name and the transparency value indicates the timestep. We see at timestep 0, all weights are the same, and as the levels progress, we find that it can naturally detect or cluster task information, showing that the model can readily detect the current task type. We posit however that the affine policy-value decoders may not have the right inductive bias to use this information well. We conjecture that the *multiplicative interaction*, via the learnt embedding, $c$, provides the right inductive bias: allowing the model to integrate information and providing additional densely-gated conditional-compute paths that leverage task similarity where appropriate, whilst guarding against interference where tasks differ – and thus more effectively learning task conditioned behaviour.

We note that we match *state-of-the-art* performance on this domain which previously used the PopArt Method (Hessel et al., 2019) (both achieve 73% normalised human score), with a simpler method. PopArt involves having task-specific value functions and using adaptive gradient normalisation (to limit interference between different reward scales). Our method also reduces the number of extra hyper-parameters needed to zero. As such, PopArt solves an orthogonal problem to the one considered here and we posit that these methods can be combined in future work. We also leave as future work the analysis of whether such hyper-value functions are able to implicitly learn the different reward scales that are explictly parameterised in methods like PopArt.

## 7 LATENT VARIABLE MODELS WITH MULTIPLICATIVE DECODERS

We previously demonstrated the ability of multiplicative interactions to extract and efficiently combine contextual information. We next explore the paradigm where the two streams of information being combined refer to *semantically* different features. Specifically, we investigate how contextual latent variables can be better integrated into neural decoders. We consider neural processes for few-shot regression. Briefly, neural processes (Garnelo et al., 2018b;a) are a neural analogue to Gaussian Processes. For few shot regression, they work by predicting a function value $\mathbf{y}^*$ at new observations $\mathbf{x}^*$ having observed previous values $(\mathbf{x}_i, \mathbf{y}_i)$ (referred to as contexts) of the same function. As opposed to training a single predictor on the $(\mathbf{x}_i, \mathbf{y}_i)$, Neural Processes learn to infer a distribution over functions that are consistent with the observations collected so far.

This is achieved by embedding context points $(\mathbf{x}_i, \mathbf{y}_i)$ individually with an *encoder* network, and then taking the mean of these embeddings. This gives latent variables $\mathbf{z}$ that are a representation of the function that maps $\mathbf{x}$ to $\mathbf{y}$, i.e. $\mathbf{y} = f(\mathbf{x}, \mathbf{z})$. A new data point $\mathbf{x}^*$ is mapped to $\mathbf{y}^*$ by passing $[\mathbf{z}; \mathbf{x}^*]$ through a *decoder* network.

We aim to increase the expressivity of the decoder by improving the conditioning on $\mathbf{z}$. The standard approach is to concatenate $\mathbf{x}$ and $\mathbf{z}$ (denoted as $\text{MLP}([\mathbf{x}; \mathbf{z}])$) leading to a purely additive relationship.

Instead, we replace the final layer of the MLP decoder with the multiplicative form $\mathcal{M}(\mathbf{x}, \mathbf{z})$. As an additional baseline, we consider skip connections between the latent variable and each layer of the decoder (denoted Skip MLP($[\mathbf{x}; \mathbf{z}]$) Dieng et al. (2018)), as a means to avoid latent variable collapse. We apply all methods to the regression task on function draws from a GP prior Garnelo et al. (2018b) and summarize results in Figure 5 a). The results show that multiplicative forms are able to better condition on latent information compared to the baseline MLPs. Further experimental details are provided in the appendix F.

## 8 MULTIPLICATIVE EMBEDDINGS FOR LANGUAGE MODELLING

Finally, we consider word-level language modelling with recurrent models. At each time-step, the network outputs a prediction about the next-word in the sequence, given the current generated word (ground truth at training) and its recurrent state. A standard architecture is to project one-hot word vectors $\mathbf{x}_t$ to input embeddings $\mathbf{z}_t^i = \mathbf{W}\mathbf{x}_t$, followed by an LSTM with output $\mathbf{h}_t$. We then produce our predicted output embedding $\mathbf{z}_{t+1}^o = \mathbf{W}_2\mathbf{h}_t\mathbf{x}_t + \mathbf{b}$ and output $\mathbf{y}_{t+1} = \text{softmax}(\mathbf{z}_{t+1}^o\mathbf{W}^T + b_2)$ where the embedding weights $\mathbf{W}$ are tied.

We posit that computing embeddings with multiplicative interactions instead will allow the model to better take its recurrent context into account. We thus compute the output embedding as follows: $\mathbf{c} = \text{relu}(\mathbf{W}_3\mathbf{h}_t + \mathbf{b})$ and $\mathbf{z}_{t+1}^o = \mathcal{M}(\mathbf{c}^T, \mathbf{h}_t)$. Instead of being quadratic in $\mathbf{h}_t$ directly we use this context vector $\mathbf{c}$ defined above. This serves two purposes: firstly, we introduce an additional on-linear pathway in the network and secondly, we have $\dim(\mathbf{c}) \ll \dim(\mathbf{h}_t)$ which allows us to drastically cut down the parameter count (for example, we have an LSTM output size of 2048, but a context size of only 32). This is an alternative to having a diagonal approximation.

We can similarly also have a multiplicative input embedding to the LSTM such that $\mathbf{z}_t^i$ also integrates recurrent information from $\mathbf{h}_{t-1}$. We could define $\mathbf{z}_t^i$ analogous to above: $\mathbf{z}_{t+1}' = \mathcal{M}(\mathbf{z}_t^i, \mathbf{h}_{t-1}) = \mathbf{z}_t^{i^T}\mathbb{W}''\mathbf{h}_{t-1} + \mathbf{z}^T\mathbf{U} + \mathbf{V}\mathbf{h}_{t-1} + \mathbf{b}$. This equation is in fact very similar to the expression used to generate the gates and candidate cell of the LSTM: the last three terms are identical. A diagonal form of the first term has been used in used inside multiplicative RNNs and LSTMs (Sutskever et al., 2011; Krause et al., 2016).

We report results (in Table 1) on running this model on Wikitext-103. For multiplicative output embeddings we use the 3D form (with the low-rank bottleneck as described above) and we use a diagonal form for the input embeddings when combining the approaches. The rest of architectural choices and hyper-parameters are reported in the appendix. We find that adding multiplicative decoder (output) embeddings provides a boost in performance and further adding input embeddings increases these gains. We ascribe this to the ability of the embeddings to now be markedly changed by context and allowing better integration of information by the inductive bias in these interactions.

Table 1: Word-level perplexity on WikiText-103

| Model | Valid | Test | No. Params |
|---|---|---|---|
| LSTM Rae et al. (2018) | 34.1 | 34.3 | 88M |
| Gated CNN Dauphin et al. (2017) | - | 37.2 | - |
| RMC Santoro et al. (2018) | 30.8 | 31.6 | - |
| Trellis Networks Bai et al. (2019) | - | 30.35 | 180M |
| TransformerXL Dai et al. (2018) | **17.7** | **18.3** | **257M** |
| LSTM (ours) | 34.7 | 36.7 | 88M |
| LSTM + MultDec | 31.7 | 33.7 | 105M |
| LSTM + MultEncDec | **28.9** | **30.3** | **110M** |

We note that we have competitive results using only a single-layer LSTM as our base model and far fewer parameters overall. Our intuition is that using such embeddings is orthognal to most of the other recent advances proposed and can thus be stacked on top of them. We leave as future work the integration of these ideas with Transformer based models.

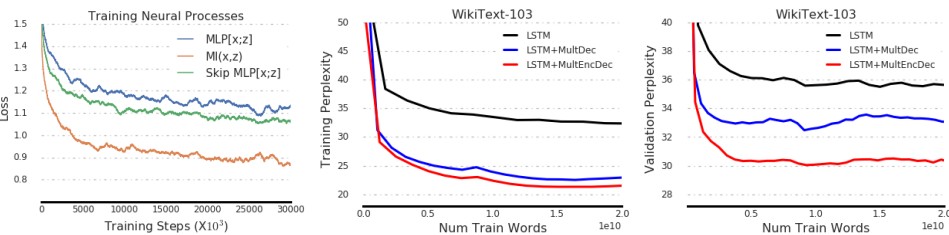

Figure 5: Results on Neural Processes and language modelling on WikiText-103.

## 9 CONCLUSION AND FUTURE WORK

In this work we considered *multiplicative interactions* and various formulations thereof, connecting them to a variety of architectures, both older and modern, such as Hypernetworks, multplicative LSTMs or gating methods. We hypothesise that the ability of such networks to better represent a broader range of algorithmic primitives (e.g. conditional-statements or inner products) allows them to better integrate contextual or task-conditional information to fuse multiple stream of data. We first tested empirically this hypothesis in two controlled settings, in order to minimize the effect of confounding factors. We further show that we could match state-of-the-art methods on multiple domains with only LSTMs and multiplicative units. While we do not necessarily advocate for a *specific* instance of the above methods, we hope that this work leads to a broader understanding and consideration of such methods by practitioners, and in some cases replacing the standard practice of concatenation when using conditioning, contextual inputs, or additional sources of information.

We believe there are many ways to explore this space of ideas more broadly, for instance looking at: the role of various approximations to these methods; ways to make their implementations more efficient; and their application to newer domains. Finally, while attention models use some of these multiplicative interactions, we hope that applying some of the lessons from this work (such as higher order interactions) will allow even greater integration of information in attention systems.

### ACKNOWLEDGEMENTS

The authors would like to thank Karen Simonyan and Sander Dieleman for their inputs and comments on the experiments as well as early drafts of the paper. We'd also like to thank Ali Razavi, Pablo Sprechmann, Alex Pritzel and Erich Elsen for insightful discussions around such multiplicative models and their applications.

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

## A EXPRESSIVITY OF THE MODEL

**Theorem 1.** *Let $\mathcal{H}_{mlp}$ denote the hypotheses space of standard MLPs with ReLU activation function, and let $\mathcal{H}_{mu}$ denote the hypotheses space of analogous networks, but with each linear layer replaced with multiplicative layer, then we have $\mathcal{H}_{mlp} \subsetneq \mathcal{H}_{mu}$.*

*Proof.* Inclusion comes directly from the fact that if we split input into arbitrary parts $[\mathbf{x}; \mathbf{z}]$ we get:

$$\mathbf{x}^T \mathbf{W} \mathbf{z} + \mathbf{x}^T \mathbf{B} + \mathbf{V} \mathbf{z} + \mathbf{c} = \mathbf{x}^T \mathbf{W} \mathbf{z} + [\mathbf{B}^T; \mathbf{V}]^T [\mathbf{x}; \mathbf{z}] + \mathbf{c} = \mathbf{x}^T \mathbf{W} \mathbf{z} + \mathbf{A}[\mathbf{x}; \mathbf{z}] + \mathbf{c},$$

which proves that $\mathcal{H}_{mlp} \subset \mathcal{H}_{mu}$. Thus, the only aspect of the theorm left to prove is that the inclusion is strict. Let us consider a 1D function $x \to x^2$, and for simplicity let $x = z$ (a domain where context equals input). A single layer MLP with a single multiplicative unit can represent this function exactly, by using $A = 0$ and $W = I$, as then we obtain $x^T W x = x^T x = \|x\|^2$. Since our function is positive, it is not affecting the multiplicative network output. For a regular MLP, let us first notice that we need at least one hidden layer, as otherwise MLP is a linear function and $f$ is not. Lets denote by $\mathbf{V}, \mathbf{c}$ and $\mathbf{w}, \mathbf{b}$ weights and biases of second, and first layers respectively. Then we have to satisfy

$$f(x) = g(\mathbf{V}^T \max(0, \mathbf{w}x + \mathbf{b}) + \mathbf{c}) = x^2,$$

where $g$ is transformation represented by all higher layers of the MLP (in particular if there are just 2 layers, then $g(x) = x$). Note that RHS is differentiable everywhere, while LHS is differentiable iff for each $i$ and for each $x$ we have $w_i x + b_i \neq 0$ (or $f$ is independent from $x$, which $x^2$ does not satisfy). However, this is impossible, as if $w_i \neq 0$, then we can always pick $x = -b_i/w_i$, and if all $w_i = 0$, then $f(x) = g(c) \neq x^2$, leading to a contradiction. $\square$

**Theorem 2.** *Let $\mathcal{H}_\sigma$ denote the hypotheses space of standard MLPs with $\sigma$ activation function, and $\mathcal{H}_w$ analogous set, where some activations are replaced with Weiestrass function $f_w$. Then we have $\mathcal{H}_{relu} \subsetneq \mathcal{H}_w$, and $\mathcal{H}_\sigma \subsetneq \mathcal{H}_w$, for any $\sigma$ that is differentiable everywhere.*

*Proof.* Inclusion comes directly from the fact that only some activations are replaced, and in particular we can always replace none, thus leading to equality of hypotheses classes. To show that the inclusion is strict lets consider a Weierstrass function itself $f(x) = \sigma_w(x)$. We definitely have $f \in \mathcal{H}_w$ as we can define 1 hidden layer network, with one hidden neuron and all the weights set to 1, and all biases to 0. Now, relu networks are piece-wise linear while the Weierstrass function is nowhere differentiable Weierstrass (1895) and thus not piece-wise linear. Similarly, network with an activation that is differentiable everywhere (e.g. sigmoid or tanh) is everywhere differentiable wrt. inputs, while Weierstrass function – nowhere Weierstrass (1895). $\square$

## B SIMPLE IMPLEMENTATION OF MI LAYER

We use Tensorflow and Sonnet Reynolds et al. (2017) for all our model implementations. The example below is a simple code snippet for adding a multiplicative layer to any model, using the Sonnet framework as an example.

```python
# Simple python code for MI Layers
import sonnet as snt
import tensorflow as tf

# A standard linear layer
# B is the batch size
# E is the input size
# C is the context size
x = ... # input of size [B, E]
z = ... # context of size [B, C]
xz = tf.concat([x,z], 1)
y = snt.Linear(output_size)(xz)

# Instead, we generate a W and b
# This defines an implicit 3D weight tensor
```

```
W = snt.Linear(output_size * input_size)(z)
b = snt.Linear(output_size)(z)

# Reshape to the correct shape
# Note: we have B weight matrices
# i.e. one per batch element
W = tf.reshape(W, [B, input_size, output_size])

# Output
y = tf.matmul(x, W) + b
```

## C   DETAILS FOR SIMPLE FUNCTION EXPERIMENTS

For the experiments modeling the simple two-input functions, we consider MLP and MI models and plot number of parameters against the input hidden size. The two models are

- MLP: linear($size$), relu, linear($output\_size$)
- MI Network: MI($size$), linear($output\_size$).

Here $size$ is the largest value such that the network has not more than N variables and we sweep over $[1, 2, 5, 10, 15, 20, 30, 40, 50, 60, 80, 100, 120, 140, 160, 180, 200]$. We sweep over learning rates 0.1, 0.001, 0.0001 and pick the best result. Models are trained using Adam optimiser for 6,000 steps using Mean Squared Error loss (MSE) on mini-batches of size 100 sampled from a standard Gaussian. The reported error is based on 10,000 samples from the same distribution to minimize the estimation variance.

## D   DETAILS FOR TOY REGRESSION EXPERIMENTS

In multitask toy regression we coinsider MLP, MI and multiheaded models. For $x \in \mathbb{R}$ (input to the task) and $z$ (represented as one-hot encoding of a task ID) we use:

- Conditional MLP: concat(x, linear(20)(z)), linear(30), relu, linear(20), relu, linear(1)
- Conditional MI: MI( [linear(30), relu, linear(20), relu](x), linear(20)(z) )
- Multiheaded MLP: linear(20), relu, linear(30), relu, linear(1); where the last linear is separate per task

Each of these models is trained in a multitask setup, where we first sample $T$ tasks (20, 40 or 60). Half of these tasks involve fitting an affine function $ax + b$, where both $a$ and $b$ are sampled from uniform distribution over $[0, 1]$; and half represent scaled sine waves, $a\sin(10x) + b$ with $a$ and $b$ sampled in the same way. Training is performed with Adam optimiser (with learning rate $3e - 3$), on mini batches of size 50 per task (so the total batch size is $50T$). Models are trained for 10,000 steps with Mean Squared Error loss (MSE), and the logarithm of the training MSE is reported for analysis. Each training is repeated 60 times to minimise the variance coming from the stochasticity.

## E   DETAILS OF MULTITASK DEEPMIND LAB TRAINING

We train multi-task on 30 DeepMind lab levels Beattie et al. (2016) concurrently using 5 actors per task and a multi-gpu learner with 4 GPUs. We follow exactly the model and hyper-parameters used in the Impala architecture Espeholt et al. (2018). Our models are all trained with population based training (PBT) Jaderberg et al. (2017) and we show below the average over three populations with 24 independent learners each for both the baseline and our best method. We train all models for 10 billion frames of data across all levels. The human normalised score is calculated independently for each level and capped at 100 (i.e $score = min(score, 100)$)

The architecture is as follows:

- Conv2D: 16ch, 8x8 kernels, stride=4
- ReLU
- Conv2D: 32ch 4x4 kernel,stride=2
- ReLU
- Linear layer with output size 256
- ReLU
- Concatenation with one hot encoded last action and last reward and language instruction
- LSTM (256 hidden units)
- Policy = Linear layer or multiplicative interaction followed by a softmax
- Value function = Linear layer or multiplicative intereaction

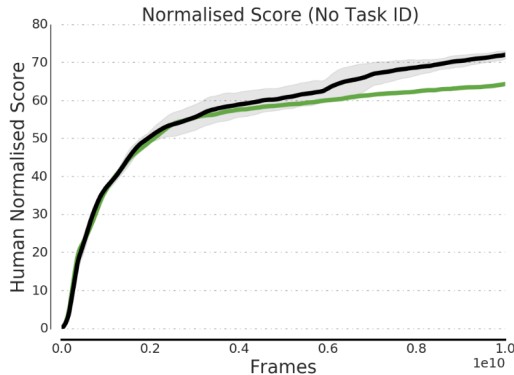

## F    DETAILS OF NEURAL PROCESS EXPERIMENTS

Inspired by the experiments in Garnelo et al. (2018b), we apply Neural Processes to functions drawn for a Gaussian Process prior with an exponentiated quadratic kernel $k(x, x') = \sigma_f^2 \exp(-\frac{1}{2}(x - x')^2/l^2)$ with fixed $\sigma^f = 1.0$ and, importantly, a random $l \sim U[0.1, 10.0]$ for each draw, resulting in a broad distribution of functions. We also add Gaussian observation noise, i.e. $y \sim \mathcal{N}(f, \sigma_n^2)$ and set $\sigma_n = 0.02$. In all experiments, we provide a deterministic transformation $h$ of the context in addition to the latent variable $z$, using separate encoder networks for each. For both SKIP MLP and the proposed MI, we concatenate $h$ and $z$ first, i.e. we use SKIP MLP$[x; \text{concat}(z, h)]$ and MI$(x, \text{concat}(z, h))$ writing concat() to denote concatenation. A more detailed discussion on adding an the benefits of adding an additional deterministic path in provided in Kim et al. (2019).

The determinsitic encoder used to obtain $h$ is a deep MLP with 6 layers of 128 units each. The latent encoder used for $z$ consists of 3 hidden layers of 128 units, parameterising mean and standard deviation of a 64-dimensional Gaussian distributed latent variable. The decoder network used to predict on new target inputs $x^*$ consists of 4 layers of 128 units. We use relu activiatons throughout all components of the network. The network is trained until convergence with Adam, using a learning rate of 0.0001 and absolute value gradient clipping with a threshold of 10.0.

## G    LANGUAGE MODELLING

We train a single layer LSTM of hidden size 2048 hidden units. The input to the LSTM is an embedding of size 256 and then output the LSTM is projected down to 256 with a single linear layer for the baseline. The input and output embedding-word matrices are tied. We use a training sequence length of 128. For the multiplicative model, the output embedding is calculated with an MI layer who's context input $c$ is generated by a linear layer with output size 32 followed by a relu. We use dropout rate of 0.3 and a learning rate of 0.001 for all models. All models are trained with the Adam optimiser.

For the multiplicative encoder we use a diagonal form of the $\mathcal{M}(.)$ layer and for the multiplicative deocder we use the full $\mathcal{M}(.)$ form (with the bottleneck described above). This amounts to adding about 20M parameters which is the same as adding 1000 hidden units. We get 6 perplexity improvement in performance whereas naively adding 1000 hidden units only gave us an improvement of 1 perplexity. Further the parameter count could be drastically reduced by considering diagonal or low rank approximations, but we do not specifically optimise for this in this work.

