# OpenReview forum: "Multiplicative Interactions and Where to Find Them"
_ICLR.cc/2020/Conference — Accept (Poster)_

### Official Review · AnonReviewer2 · 2019-10-23
**Official Blind Review #2**

**Rating:** 6

**Review:**

This paper presents multiplicative interaction as a unified characterization for representing commonly used model architecture design components (e.g. gating, attention layers and hypernetworks). Multiplicative interactions can be viewed as an effective way of integrating contextual information in a network. Through a series of thorough empirical experiments, this paper demonstrates superior performance on a variety of tasks (RL, sequence modeling) when a such multiplicative interaction module is incorporated.

The framework seems applicable for learning tasks where a latent variable or context embedding presents. And the paper hypothesizes that multiplicative interactions can help introduce desirable inductive biases, therefore leading to improved generalization performance. However, the theoretical intuition and justification for such hypothesis is missing, which weakens the contribution of the paper.

In Table 1, the model LSTM with Multiplicative Decoder has more parameters (105M) than the vanilla LSTM in comparison (88M). It’d be good to provide a more fair comparison by slightly increasing the capacity of baseline LSTM model (e.g., using larger output dimension to match the capacity). This will rule out the cofounding factor that the improved performance is due to the algorithm instead of increased model capacity.

In Section 8, it’d be convincing if the authors can also report results for replacing default LSTM cell with multiplicative interactions. In paragraph 3, the authors only discussed the feasibility conceptually, without providing experimental results to support this argument.


**Experience Assessment:**

I do not know much about this area.

**Review Assessment: Checking Correctness Of Derivations And Theory:**

N/A

**Review Assessment: Checking Correctness Of Experiments:**

I carefully checked the experiments.

**Review Assessment: Thoroughness In Paper Reading:**

I read the paper thoroughly.

---

> ### Author Response · Authors · 2019-11-08
> **Author Response**
>
> We thank the reviewer for their comments and suggestions for further experiments.
>
> Re fair comparison: This is absolutely valid and we have now re-run LSTMs upto 3000 hidden units (with 108M parameters). These are still running but what we see is that the bigger LSTMs provide some data-efficiency gains (in terms of plateauing at a lower number of iterations), but provide zero to very small (0.1) gain in perplexity over the 2048 hidden size baseline LSTM (88M parameters). Thus even controlling for parameters we can see there is a large gain in performance using such multiplicative interactions and believe this rules out the confounding factor you mention. We will add these results in the appendix once they have finished running completely.
>
> Re default LSTM cell: Would you able to clarify what you mean by “replacing” the cell? What we found is that adding in additional multiplicative interactions in this way described, provides a boost in performance. The process we refer to in paragraph 3 is what is called “LSTM+MultEncDec” in the table.

---

### Official Review · AnonReviewer3 · 2019-10-25
**Official Blind Review #3**

**Rating:** 8

**Review:**

The paper explores different types of multiplicative interactions. This allows understanding of both efficacy of multiplicative interaction (i.e., MI vs MLP), and the common between multiplicative-type models (e.g., hypernetworks, FiLM, gating,  attention etc). The authors also find MI models able to achieve a state-of-the-art performance on language modeling (WikiText-103) and reinforcement learning problems(DMLAB-30), along with several toy examples.

Strengths:
*The discussed MI subject is important research area, the paper presents the vast related work, proven by the fact MI techniques appears in many recent models.
* The authors  did comprehensive evaluation to show importance and usefulness of multiplicative interactions. A  toy regression experiment to show superiority of MI vs MLP . A Reinforcement learning task, i.e DMLAB-30 on par with the state-of-the-art model, but with simple model (i.e., less parameters). A sequence prediction with an alternative embedding technique that improves both accuracy and number of needed parameters.
* The paper is of high quality: well organized, clear, with good supporting figures.

Weaknesses:
* I suggest a better explanation how the suggested models compare to state-of-the-art models.  Currently, it is hard to assess the impact. For instance, proposed model alternate existing baselines, such as PopArt, or is completely novel?
* Although mentioned, it's not the focus of this work, the paper should have discussed attention models more.  Specifically recent years multimodal attention relied on multiplicative interactions. Therefore, at least in this domain multiplicative interactions are not "under-appreciated". Relevant papers: [1, 2, 3, 4] - advancement of multiplicative interactions over the years, [5] - introduce co-attention, [6] - introduce higher-order interactions (between three vectors), [7] - introduce bilinear attention .

To conclude, multiplicative interactions are extremely important, and I find the paper exploration useful. In addition,  the paper is of high quality, and with satisfied experiments to prove their claims. I do suggest a better discussion about multimodal attention networks, which are relevant examples.

[1] - Ask, Attend and Answer: Exploring Question-Guided Spatial Attention for Visual Question Answering; Xu et al.
[2] - Multimodal Compact Bilinear Pooling for Visual Question Answering and Visual Grounding; Fukui et al.
[3] - Hadamard Product for Low-rank Bilinear Pooling; Kim et al.
[4] - Multi-modal Factorized Bilinear Pooling with Co-Attention Learning for Visual Question Answering; Yu et al.
[5] - Hierarchical Question-Image Co-Attention for Visual Question Answering; Lu et al.
[6] - High-Order Attention Models for Visual Question Answering; Schwartz et al.
[7]  - Bilinear Attention Networks; Kim et al.


**Experience Assessment:**

I have published one or two papers in this area.

**Review Assessment: Checking Correctness Of Derivations And Theory:**

I assessed the sensibility of the derivations and theory.

**Review Assessment: Checking Correctness Of Experiments:**

I assessed the sensibility of the experiments.

**Review Assessment: Thoroughness In Paper Reading:**

I read the paper thoroughly.

---

> ### Author Response · Authors · 2019-11-12
> **Author Response**
>
> We thank the reviewer for their detailed comments and links to the relevant multimodal attention literature. We have revised the paper with an increased discussion of these papers. Our specific reasoning for not including more discussion around attention networks originally was that their efficacy is often attributed not just to multiplicative interactions but also their ability to make connections across vast temporal horizons. We thus felt a more detailed study to tease apart these two functions was needed, which was out-of-scope for this work. However the paper links are certainly relevant, as we already some works in  VQA and the works linked are able to combine both attention and such bilinear systems. We have thus expanded the overall related work.
>
> We have also expanded the discussion around PopArt and other baselines. We feel that the performance gain from PopArt is orthogonal as it addresses a different underlying issue: namely the difference in reward/loss scales in RL. As such these two methods could be combined.

---

### Official Review · AnonReviewer1 · 2019-11-07
**Official Blind Review #1**

**Rating:** 6

**Review:**

This paper takes a component of neural networks that is sometimes, but not always used — multiplicative interactions — and goes into depth in analyzing and discussing the different ways in which computed features may interact multiplicatively. The analysis and math are clear, and the main points are made with neither too many nor too few equations. Authors cast several other approaches (gating, attention, Hypernets) within the same formulation, which while not being groundbreaking or overly novel is nice to see compiled into this particular form.

Decision: weak accept.

Pros:
 - I can imagine any researcher getting into the field and interested in this sort of thing reading this paper as a canonical, concise, and moderately thorough intro to the concept.
 - Beautiful drawing

Cons:
 - Experiments are a little light. To make the paper more thorough, it would be nice to see cases where multiplicative interactions are *not* needed, e.g. if one sticks multiplicative interactions in a network trained to classify MNIST or CIFAR-10, do they help at all? If not, why not?
 - Further ablation studies would also benefit the paper. For example, in the language modeling section, how would the results change with varying bottleneck size? With a completely diagonal approximation? Finally, LSTMs already have multiplicative interactions in them (via gating, as mentioned in the paper). Would the proposed form of multiplicative interaction obviate the need for LSTM gating all together?

With additional experiments this paper could be a strong accept and recommended reading for many first year grad students. As is, it’s probably still worth accepting.


**Experience Assessment:**

I have published in this field for several years.

**Review Assessment: Checking Correctness Of Derivations And Theory:**

I assessed the sensibility of the derivations and theory.

**Review Assessment: Checking Correctness Of Experiments:**

I assessed the sensibility of the experiments.

**Review Assessment: Thoroughness In Paper Reading:**

I read the paper thoroughly.

---

> ### Author Response · Authors · 2019-11-08
> **Author Response**
>
> We thank the reviewer for their detailed feedback and comments on the paper. Part of our aim in writing this was to provide a concise study into multiplicative interactions and we will definitely take on board the comments to make this case stronger.
>
> The point about where multiplicative interactions are *not* needed is very valid. We ran some preliminary experiments which showed that they were not out of the box useful in such classification domains. Our intuition was that this was not a domain where different contextual streams were being combined per-se, and that this probably explained the lack of improvement. As a large part of the study considers combining such streams of information, we thought these experiments were out-of-scope and didn’t include them. We respectfully disagree with the comment that the experiments are light: we believe the results across large scale RL, and LM, combined with improvements in neural processes and the exploratory toy domains provide diverse and compelling evidence that these methods are widely applicable.
>
> On the point about bottle-neck size: we will make this clearer in the paper. What we found was that bottleneck was not too important in determining results. What was true was that below a certain size the network could not learn but there was no large improvement in increasing it above 64 (ie 2048 —> 64).
>
> Re LSTM gating: We don’t think the proposed form obviates the need for LSTM gating — as the gating itself can be seen as a diagonal version of the proposed form with added non-linearities. We think what we do show is there is not *enough* such gating inside an LSTM. Specifically, the added multiplicative interactions between x and h_t improve performance significantly.

---

### Decision · Program_Chairs · 2019-12-19

**Decision:**

Accept (Poster)

**Comment:**

This paper provides a unifying perspective regarding a variety of popular DNN architectures in terms of the inclusion of multiplicative interaction layers.  Such layers increase the representational power of conventional linear layers, which the paper argues can induce a useful inductive bias in practical scenarios such as when multiple streams of information are fused.  Empirical support is provided to validate these claims and showcase the potential of multiplicative interactions in occupying broader practical roles.

All reviewers agreed to accept this paper, although some concerns were raised in terms of novelty, clarity, and the relationship with state-of-the-art models.  However, the author rebuttal and updated revision are adequate, and I believe that this paper should be accepted.